# *Cronobacter* spp. Isolated from Quick-Frozen Foods in China: Incidence, Genetic Characteristics, and Antibiotic Resistance

**DOI:** 10.3390/foods12163019

**Published:** 2023-08-11

**Authors:** Qi Li, Chengsi Li, Ling Chen, Zhihe Cai, Shi Wu, Qihui Gu, Youxiong Zhang, Xianhu Wei, Jumei Zhang, Xiaojuan Yang, Shuhong Zhang, Qinghua Ye, Qingping Wu

**Affiliations:** Key Laboratory of Agricultural Microbiomics and Precision Application (MARA), Guangdong Provincial Key Laboratory of Microbial Safety and Health, State Key Laboratory of Applied Microbiology Southern China, Institute of Microbiology, Guangdong Academy of Sciences, Guangzhou 510070, China; liqiouc@sina.com (Q.L.); lichengsi@gdim.cn (C.L.); chenling@gdim.cn (L.C.); caizhihe1971@163.com (Z.C.); wushiloveyou@126.com (S.W.); guqh888@163.com (Q.G.); zyx0713141061@126.com (Y.Z.); wxhu7508@163.com (X.W.); zhangjm128@126.com (J.Z.); youngxj@126.com (X.Y.); zhangshh2001@163.com (S.Z.); yeqinghua2002@163.com (Q.Y.)

**Keywords:** pathogenic *Cronobacter*, quick-frozen foods, prevalence, genetic diversity, multiple antimicrobial resistance

## Abstract

*Cronobacter* spp. are emerging foodborne pathogens that cause severe diseases. However, information on *Cronobacter* contamination in quick-frozen foods in China is limited. Therefore, we studied the prevalence, molecular characterization, and antimicrobial susceptibility of *Cronobacter* in 576 quick-frozen food samples collected from 39 cities in China. *Cronobacter* spp. were found in 18.75% (108/576) of the samples, and the contamination degree of the total positive samples was 5.82 MPN/g. The contamination level of frozen flour product samples was high (44.34%). Among 154 isolates, 109 were *C. sakazakii*, and the main serotype was *C. sakazakii* O1 (44/154). Additionally, 11 serotypes existed among four species. Eighty-five sequence types (STs), including 22 novel ones, were assigned, indicating a relatively high genetic diversity of the *Cronobacter* in this food type. Pathogenic ST148, ST7, and ST1 were the main STs in this study. ST4, epidemiologically related to neonatal meningitis, was also identified. All strains were sensitive to cefepime, tobramycin, ciprofloxacin, and imipenem, in which the resistance to cephalothin was the highest (64.94%).Two isolates exhibited multidrug resistance to five and seven antimicrobial agents, respectively. In conclusion, these findings suggest that the comparatively high contamination level of *Cronobacter* spp. in quick-frozen foods is a potential risk warranting public attention.

## 1. Introduction

*Cronobacter* spp. are key members of the *Enterobacteriaceae* family [1]. They are burgeoning food-related pathogens classified into seven species [2]. Most are believed to be pathogenic because of the retrospective connection of each with clinical infections in adults and infants [3,4]. Prominent symptoms, including necrotizing enterocolitis, septicemia, and brain abscess, predominately manifested in neonates and infants with high fatality rates, are usually related to *Cronobacter sakazakii* [5,6,7]. In older and immunocompromised adults, *Cronobacter malonaticus* is considered to be linked to urosepsis and bacteremia [8,9]. Studies have revealed that *Cronobacter* spp. are plant-associated [10,11]. Therefore, the genus can be detected in various foods of plant origin (such as vegetables, spices, and herbs) [12,13]. The genus has also been reported to exist in several abiotic sources (e.g., hospital equipment, household utensils, and food processing environments) [14]. Additionally, the widespread occurrence of *Cronobacter* spp. in food is pertinent to their high resistance to diverse ambient conditions, such as resistance to dry and osmotic conditions, along with high thermal tolerance [15].

Quick freezing is a processing technology that quickly freezes food within 15 min. Recently, quick-frozen foods have been chosen more often by customers globally because of simple preparation, better preservation of nutrients, and convenient storage. The supply chain for quick-frozen foods includes several steps, such as purchasing raw materials, processing, circulation, and sale. These complex procedures increase the potential risk of microbial contamination. A low temperature can inhibit the growth of most bacteria. However, some cold-resistant pathogens can survive at low temperatures [16]. Inadequate control of low temperatures or interruption of the cold chain can also promote microbial reproduction. This causes food deterioration with slime and off-flavors, threatening public health and the economy [17,18]. Thus, more stringent measures are required to eliminate the waste caused by microorganisms and monitor the safety of quick-frozen foods.

In our previous studies, we isolated several foodborne pathogens from quick-frozen foods, of which *Listeria monocytogenes* was prevalent at 44.90% [19,20,21,22,23]. These discoveries highlight the importance of investigating the microbiological profile of quick-frozen food products for safety. Although powdered infant formula is a common vehicle of *Cronobacter* [5,24], different foods have also been found to be contaminated with this bacterium [25,26,27]. Its prevalence was found to be especially high in certain types of food samples. For instance, *Cronobacter* spp. were found in cereals and dried fish, with prevalence rates of 54.00% and 55.60%, respectively [28,29]. Nevertheless, limited information is available on the microbiological quality and the genetic characteristics of *Cronobacter* in quick-frozen foods. Miranda et al. [30] reported that one frozen ravioli product was positive for *Cronobacter* spp. among five tested frozen food samples. However, it is difficult to obtain convincing results and perform an analysis of genetic diversity with limited sample numbers. People tend to be affected by quick-frozen foods contaminated by *Cronobacter* spp. as this type of food is an essential part of a common diet. Hence, a large-scale survey on *Cronobacter* spp. contamination in quick-frozen foods is urgently needed.

*Cronobacter* spp. are sensitive to most antimicrobials [31]; thus, antibiotic therapy has been demonstrated to be clinically useful in treating infections [32]. However, with the indiscriminate application of antibiotics, *Cronobacter* has evolved continuously for survival and has developed antibiotic resistance, particularly to penicillins and cephalosporins [33,34]. Bacteria have the following types of antibiotic-resistance mechanisms: (1) altered permeability of cell membranes, limiting antibiotic intake; (2) modulation of antibiotic binding targets to prevent binding of antibiotics to bacteria [35]; (3) inactivation of antibiotics through the production of catalytic enzymes; (4) elimination of antibiotics from bacteria via an efflux pump [36]. As more *Cronobacter* strains with multidrug resistance have been isolated, traditional treatments have become ineffective, resulting in increased pain and disadvantage to the patients. Thus, investigating the antimicrobial resistance profile of *Cronobacter* spp. is crucial and warranted.

In this study, we aimed to analyze a large number of samples and assess the incidence, antibiotic resistance, and genetic characterization of *Cronobacter* detected in quick-frozen foods from different cities in China. O-antigen serotyping and multilocus sequence typing (MLST) were used to investigate the molecular features of *Cronobacter* isolates. According to highly polymorphic O-antigen structures, serotyping methods based on PCR have been broadly applied to strain categorization, epidemiological assessment, and outbreak monitoring [37,38]. MLST is a highly sensitive technique developed based on analyzing seven loci (*fusA*, *gyrB*, *glnS*, *atpD*, *infB*, *gltB*, and *pps*) for classifying and genotyping pathogens. The data of 3819 *Cronobacter* strains isolated from various sources and countries, including provenance and phenotype linked to molecular typing, are recorded in the *Cronobacter* PubMLST database1. MLST has, therefore, become the most popular tool for source tracking and investigating the genetic diversity of *Cronobacter* isolates compared with database information [39,40,41]. Our results can provide a systematic record for comprehending *Cronobacter* contamination in quick-frozen foods in China and evaluating potential consumer risks.

## 2. Materials and Methods

### 2.1. Sampling

From July 2011 to June 2016, a total of 576 quick-frozen food samples were bought from different retail stores and supermarkets in 35 capital cities distributed in 21 provinces and five autonomous regions, two directly controlled municipalities, and two special administrative regions in China. (Appendix A, Appendix A). The samples were composed of 212 frozen flour products (frozen dumplings/steamed stuffed buns/wonton), 239 frozen poultry samples (frozen chicken/duck), and 125 frozen meat samples (frozen mutton/pork/beef). Each sample was placed separately in a sterile bag, labeled, and stored in cool box when transported to the laboratory. Further, the microbiological assays of samples were conducted within 24 h.

### 2.2. Quantitative Detection and Classification of Cronobacter spp.

Quantitative detection was prepared according to the National Food Safety Standards of China document GB 4789.4-2010, and the process was described in previous research [12]. In accordance with a previous report, the most probable number (MPN) was calculated [42]. The systems of API 20E (BioMérieux, Marcy-l’Étoile, France) were recommended to identify the genus. Bacteria displaying green or blue-green colonies on a chromogenic *Enterobacter sakazakii* Agar Plate were characterized as presumptive *Cronobacter*. In addition, *fusA* gene sequencing, a speciation method with high accuracy, was used [40,43]. BLASTn was used to complete the identification by uploading the sequenced results to NCBI.

### 2.3. O-Antigen Serotyping and Multilocus Sequence Typing

Genomic DNA was extracted from the tested isolates using the HiPure Bacterial DNA Kit (Huankai, Guangzhou, China). The PCR conditions of O-antigen serotyping resembled those of previous studies [23,30]. The results were characterized using agarose gel electrophoresis, and the serotypes were decided by the only targeted band. MLST was applied to molecular typing of *Cronobacter* strains, and amplification of seven house-keeping genes was operated by the primers and PCR programs recommended in the *Cronobacter* MLST database. The PCR products were subjected to bidirectional sequencing by Beijing Genomics Institute (Shenzhen, China). The DNA sequences were then uploaded to the MLST database to obtain allele profiles and STs. New alleles and STs were designated by Professor Steve Forsythe, the curator of the *Cronobacter* MLST database. BioNumerics 8.1.1 (Applied Maths, Sint-Martens-Latem, Belgium) was applied to create a minimum spanning tree to reflect the phylogenetic relationships of concatenated sequences.

### 2.4. Antimicrobial Susceptibility Analysis

Based on the instructions of the Clinical and Laboratory Standards Institute [44], the Kirby–Bauer diffusion method was used to evaluate the antimicrobial agents (AMs) susceptibilities of *Cronobacter* isolates by diluting antibiotics and analyzing the sensitivities of the disks displayed in Mueller–Hinton agar (Huankai). A total of 16 antimicrobial agents (Oxoid, Hampshire, United Kingdom) were detected in this experiment: ampicillin (AMP, 10 μg), ampicillin/sulbactam (SAM, 10 μg), cefepime (FEP, 30 μg), ceftriaxone (CRO, 30 μg), cefazolin (KZ, 30 μg), cephalothin (KF, 30 μg), gentamicin (CN, 10 μg), tobramycin (TOB, 10 μg), amikacin (AMK, 30 μg), ciprofloxacin (CIP, 5 μg), imipenem (IPM, 10 μg), trimethoprim/sulfameth-oxazole (SXT, 25 μg), aztreonam (ATM, 30 μg), amoxicillin-clavulanic acid (AMC, 30 μg), chloramphenicol (C, 30 μg), and tetracycline (TE, 30 μg). We characterized the susceptibilities of the analyzed isolates after 24 h at 37 °C by measuring the inhibition region and illuminating the diameters according to the guidelines.

### 2.5. Statistical Analysis

The SPSS software (version 22.0, SPSS, Inc., Chicago, IL, USA) was used to perform statistical analysis. The chi-square test was used to compare differences of the positive sample numbers in the three types of quick-frozen foods and the prevalence among differentcities. A *p*-value <0.05 was considered to indicate significant differences.

## 3. Results

### 3.1. Contamination Level and Isolation of Cronobacter spp. in Quick-Frozen Foods

In total, 576 quick-frozen food samples were purchased from 2011–2016 from 39 typical cities geographically spread across China (Table 1, Appendix A). Furthermore, 18.58% (107/576) of the samples tested positive for *Cronobacter* spp., and the contamination level of the total positive samples was 5.82 MPN/g. Among these samples, the level of *Cronobacter* contamination of frozen flour products was 44.34% (94/212), followed by frozen meat (4.00%) and frozen poultry (3.77%; *p* < 0.01). Overall, 93.52% (101/108) of the samples were contaminated with *Cronobacter* spp. at <10 MPN/g. Only three and four samples of frozen flour products were within 10–110 MPN/g and ≥110 MPN/g, respectively. Among frozen flour product samples, the prevalence of *Cronobacter* spp. in dumplings was 49.09% (81/165), followed by wonton (47.06%; *p* < 0.01). We had nine positive frozen poultry samples, all from chicken. Finally, the highest number of positive samples was identified in pork (13.64%; *p >* 0.05), and no *Cronobacter* species were detected in beef (0/36) in frozen meat-type products. (Appendix A). The prevalence of *Cronobacter* spp. varied in different cities, and the contamination levels in Hefei (38.46%) and Xi’an (35.71%) were significantly higher than those in other cities (*p >* 0.05). The 39 sampling sites distributed all over China were divided into seven regions (Appendix A) [12,21]. East China had the most serious contamination by *Cronobacter* (26.36%), followed by Northwest China (21.21%; Table 2). The prevalence of *Cronobacter* was the lowest in Southwest China at 12.73% (*p >* 0.05).

According to *fusA* sequencing, four species were characterized from 154 *Cronobacter* strains isolated from 107 contaminated samples (i.e., 109 *C. sakazakii* strains, 22 *C. malonaticus*, 19 *C. dublinensis*, and 4 *C. turicensi*; Table 3). Regarding these 154 *Cronobacter* isolates, 140 were from frozen flour products, nine were from frozen poultry samples, and five were from frozen meat samples.

### 3.2. Antimicrobial Susceptibility Testing

We selected 16 antibiotics from nine groups to examine the susceptibility, intermediate resistance, and resistance profiles of 154 *Cronobacter* isolates (Figure 1, Appendix A). All the strains were sensitive to cefepime, tobramycin, ciprofloxacin, and imipenem. However, they were most resistant to cephalothin, with resistance and intermediate resistance rates of 64.94% and 27.27%, respectively, followed by cefazolin, with an 11.69% intermediate resistance and a 1.30% resistance rate. Moreover, six isolates were resistant to two or more antibiotics. Five isolates were collected from frozen flour products and one from frozen poultry meat. One *C. malonaticus* strain—cro315B1—was resistant to five antibiotics (ampicillin, gentamicin, ampicillin/sulbactam, chloramphenicol, and tetracycline). The cro1892B2 (*C. sakazakii*) isolate was resistant to seven antibiotics: ampicillin, cefazolin, cephalothin, ceftriaxone, aztreonam, chloramphenicol, and tetracycline.

### 3.3. Molecular Serotype and MLST Patterns of Cronobacter Isolates

Table 3 indicates that five *C. sakazakii* serotypes were distributed in the 154 *Cronobacter* strains; specifically, O1 had 44 isolates, O2 had 23 isolates, and O3, O4, and O7 had 13, 11, and 16 isolates, respectively. Twenty-two *C. malonaticus* strains were classified into three serotypes: O1 (*n* = 5), O2 (*n* = 11), and O3 (*n* = 5). We discovered two *C. dublinensis* (O1 and O2) and one *C.turicensis* (O3) serotypes. However, cro2845B2 (*C. dublinensis*), cro517A2 (*C. malonaticus*), and *C. sakazakii* isolates of cro2239A1 and cro2244W were recognized as uncertain serotypes. The above results demonstrated that *C. sakazakii* O1 was the predominant serotype, followed by *C. sakazakii* O2. Additionally, three serotypes were detected in each sample of the seven frozen dumpling samples, and two serotypes were detected in every sample of the 20 frozen flour products.

Using MLST analysis, 85 STs, containing 22 new STs, were assigned to 154 *Cronobacter* isolates: ST148 (8/154, 5.19%) was the dominant ST, followed by ST7 and ST1, both of which had six isolates. Fifty-seven STs, with each ST in one isolate, and the residual 28 STs were separately preserved by 2–8 strains. Twenty-two novel STs were composed of seven *C. sakazakii* isolates, four *C. malonaticus* isolates, nine *C. dublinensis* isolates, and two *C.uricensis* isolates. All loci, except for *fusA*, tested new allele types, most of which were observed in *pps*, in which 11 new allele types were characterized.

A minimum spanning tree was generated to demonstrate the relationship of ST with sample sources and serotypes among the 154 isolates (Figure 2A,B). Most STs, such as ST148, ST7, ST1, and ST13, were only found in frozen flour products. In addition, ST606, ST264, and ST300 were only identified in frozen poultry samples, and ST21 and ST701 in frozen meat samples. ST73 and ST60 were observed in frozen flour products and poultry samples, and ST58 and ST917 were observed in frozen flour products and meat samples. Every serotype identified in this investigation displayed various MLST patterns. The maximum number of MLST modes was 19 in *C. sakazakii* O1, followed by *C. sakazakii* O2 (*n* = 14). Some STs only corresponded with a specific serotype; for example, ST148, ST58, and ST256 isolates only presented in *C. sakazakii* O1; ST308, ST3, and ST64 strains were unique to *C. sakazakii* O2; and ST7, ST201, and ST901 appeared in *C. malonaticus* O2 isolates. Conversely, some STs, such as ST8, ST13, and ST249, were associated with two distinctive serotypes. Thus, the MLST results could more precisely classify and indicate higher genetic diversity of quick-frozen food isolates than O-antigen serotyping.

## 4. Discussion

Despite advances in studies on the prevalence of *Cronobacter* strains in various food samples in the past few years, little information exists on the contamination level of quick-frozen foods. Thus, in the current study, a large number of quick-frozen food samples covering most regions of China were selected, making our results more persuasive and representative. We found the general prevalence of *Cronobacter* spp. to be 18.75% in our samples. This was higher than the prevalence reported in other types of food, such as dried edible mushrooms (14.80%), meat and meat products (9.18%), and aquatic products (3.90%) in China and infant cereal (17.33%), desiccated foods (15.65%), and salads (8.20%) in other countries [25,26,45,46,47,48]. The prevalence of *Cronobacter* in frozen flour products was high (44.34%), consistent with previous findings that *Cronobacter* is plant-associated pathogen and that plants may be its natural ecosystem [8]. In most cases, the contamination level of *Cronobacter* spp. in plant products was comparatively severe [48,49,50]. There may be variations in quick-frozen samples and conditions of supply chains at different places. This may explain the difference in the prevalence rate of *Cronobacter* in the seven regions in China and warrants further research. Hygiene standards in food-chain management and food manufacturing practices have to be implemented urgently, in addition to providing adequate cold-storage conditions in food processing and retail sectors in cities of China, especially in Shanghai, Hefei, and Xi’an.

Freezing is commonly used to preserve the taste and quality of food and effectively minimize the growth and reproduction of foodborne pathogens. The growth temperature of *Cronobacter* spp. is 5–47 °C. Thus, refrigerating temperature (4 °C) can inhibit the growth of *C. sakazakii* in fresh fruits [51,52]. However, we observed a relatively high degree of *Cronobacter* spp. contamination in quick-frozen foods; these foods are usually subjected to rapid freezing below 30 °C and stored and circulated below 18 °C until used. These quick-frozen foods may be contaminated at each stage of thesupply chain, or the cold chain might be interrupted, causing the products to develop a damp or frosted surface, which could lead to thermal changes resulting in the growth of pathogens. High *Cronobacter* prevalence in frozen flour products implies that these foods are probably reservoirs for and transmitters of *Cronobacter* in China. This pathogen has remarkable tolerance to desiccation, low pH, and thermal conditions [15], and these properties facilitate the long-term survival of *Cronobacter* and its persistence in foods and food-processing environments, which potentially causes cross-contamination. Therefore, improving the manufacturing hygiene standards and strictly controlling the cold-chain conditions for the safety of quick-frozen foods is crucial.

*Cronobacter* spp. infections are treated with antimicrobial agents. In the present study, most isolates were susceptible to a wide spectrum of antibiotics. The susceptibility of *Cronobacter* to gentamycin and chloramphenicol—the most widely used clinical antibiotics—was high, at 98.70% and 98.05%, respectively. These findings are consistent with those of earlier investigations of *Cronobacter* isolates from other varieties of food [25,53]. Likewise, the cephalothin-resistant *Cronobacter* isolates detected in this study are again consistent with those reported in previous studies on other food types [26,31,47]. This reveals the importance of rationally administering cephalothin in China. Moreover, compared with other food-related pathogens, *Cronobater* spp. presented a relatively low resistance to antibiotics [54,55]. Epidemiological observations of *Cronobater* spp. are scarce compared to those of other common foodborne pathogens, possibly because this genus is an emerging microbe.

Multidrugresistance to foodborne pathogens should raise our attention because it affects human health. We found that cro315B1 and cro1892B2 strains were resistant to five and seven antibiotics separately. This is possibly an alarming sign of the emergence of more multidrug-resistant *Cronobacter* isolates if antimicrobials are not carefully managed. Many multidrug-resistant *Cronobacter* isolates have shown the ability to horizontally transfer antibiotic-resistance genes among *Cronobacter* species [34]. Consequently, understanding the genetic diversity of this bacterium is important in epidemiological studies, as it plays a vital role in formulating a more efficient and targeted control strategy for *Cronobacter*.

The molecular serotypes distribution varied in different food types. For example, *C. sakazakii* O1 was dominant in the edible mushrooms, raw vegetables, and quick-frozen foods analyzed in this study [12,53]. In contrast, ready-to-eat foods and infant nourishments are commonly contaminated with *C. sakazakii* O2 [42,56]. Considering the clinical implication, *C. sakazakii* O1, O4, and O2 and *C. malonaticus* O1 are associated with human diseases [57]. Altogether, 83 isolates of 154 *Cronobacter* strains (53.90%) belonged to these virulent serotypes, suggesting the hazards of quick-frozen foods and potential risks to the public.

Similarly, the type of food sample also affected the predominant ST; ST148, the leading ST in this survey, was also dominant in spices and cereals [38]. ST7 and ST1 were the second most common STs in quick-frozen samples and are the major STs in aquatic products and human stool in the clinic [26]. *C. sakazakii* ST4 and *C. malonaticus* ST7 have been isolated from the patient’s sputum and throat samples. ST4 is the primary *Cronobacter* neonatal meningitis pathovar [58]. ST7 is linked to adult infections [40]. In this study, three of four ST4 and all ST7 isolates were obtained from frozen flour products. Furthermore, 35.44% (28/79) of STs identified in this study have been isolated from clinical sources in various countries, according to the PubMLST database. Our results indicate that quick-frozen foods contaminated with *Cronobacter* spp., especially frozen flour products, pose a significant risk to consumers’ health. Thus, more stringent controls preventing contamination of quick-frozen foods with *Cronobacter* spp. are required in China.

## 5. Conclusions

We report the results of a large-scale and long-term survey of *Cronobacter* contamination in quick-frozen foods in China. The high percentage of clinically relevant serotypes and the detection of pathogenic STs and multidrug-resistant isolates in this study imply a potentially severe risk to the public. Therefore, tracking the source of contamination, characterizing the transmission routes of *Cronobacter* in quick-frozen foods, and analyzing the relationship of *Cronobacter* in different samples from the environment, clinical settings, and foods should be the subjects of future studies.

## Figures and Tables

**Figure 1 foods-12-03019-f001:**
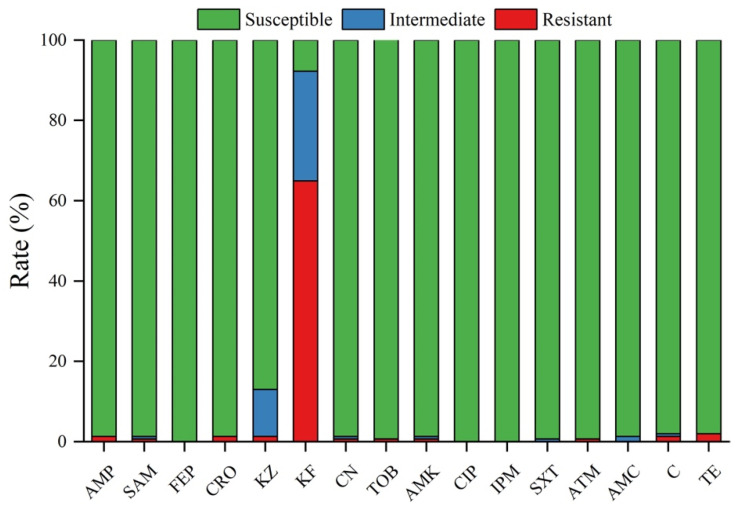
Antimicrobial susceptibility of *Cronobacter* spp. identified in this study. The susceptible, intermediate resistant, and resistant rates of 154 strains to 16 different antibiotics are represented by the green, blue, and red bars, respectively.

**Figure 2 foods-12-03019-f002:**
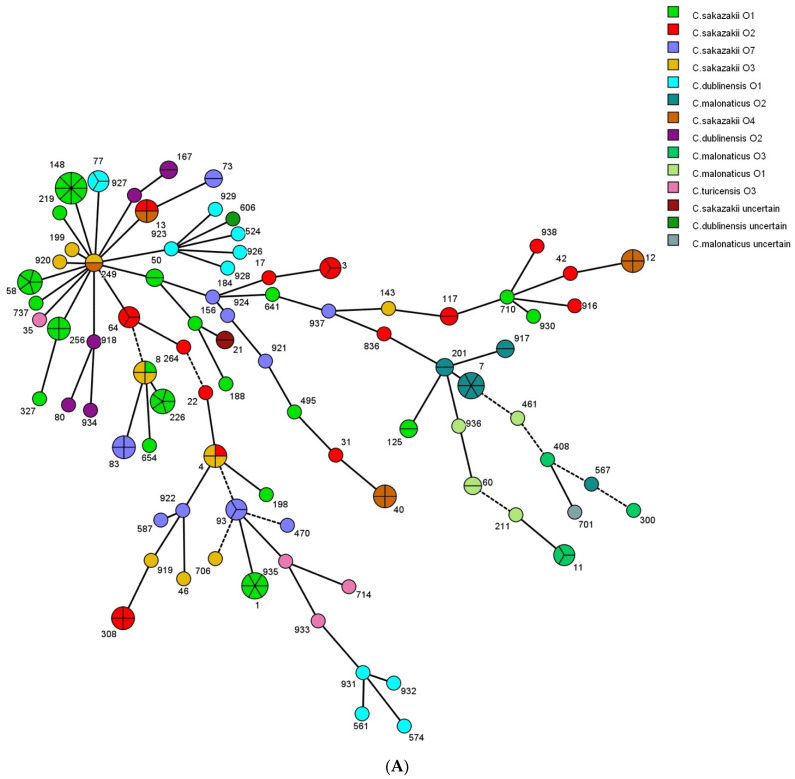
Clustering relationship between *Cronobacter* spp. isolated in this study. BioNumerics 8.1.1 software was used to produce a minimum spanning tree relying on MLST results of the 154 *Cronobacter* isolates obtained from 576 quick-frozen food samples. Each ST was represented by one circle and the ST numbers were displayed next to the corresponding circles. The circle diameter was associated with isolate number of corresponding ST. The colors within the circles were the symbols of the relevant serotypes (**A**) and sources (**B**).

**Table 1 foods-12-03019-t001:** The contamination level of *Cronobacter* spp. in quick-frozen foods analyzed in this study.

Sample	No. of Samples	No. (%) of Positive Samples (*p* < 0.01)		No. of Positive Samples by Quantitative Methods by MPN/g Range			Positive Sample Contamination Level (MPN/g)
				MPN < 10	10 ≤ MPN < 110	110 ≤ MPN	
Frozen flour products	212	94	44.34%	87	3	4	6.63
Frozen poultry	239	9	3.77%	9	0	0	0.42
Frozen meat	125	5	4.00%	5	0	0	0.22
Total	576	108	18.75%	101	3	4	5.82

**Table 2 foods-12-03019-t002:** The prevalence rate of *Cronobacter* spp. in quick-frozen food samples in seven geographical regions of China.

Region	No. of Samples	No. of Positive Samples	Prevalence Rate (%) (*p >* 0.05)
South China	207	35	16.90%
East China	110	29	26.36%
Central China	41	7	17.07%
North China	55	9	16.36%
Northeast China	42	7	16.67%
Northwest China	66	14	21.21%
Southwest China	55	7	12.73%

**Table 3 foods-12-03019-t003:** Isolates, serotypes, and MLST patterns of 154 *Cronobacter* strains isolated in this study. House-keeping genes and STs marked by * are new.

Isolates (Total)	Serotypes	Total	Type of Samples			MLST Allelic Type							
			Frozen Flour Products	Frozen Poultry	Frozen Meat	*atpD*	*fusA*	*glnS*	*gltB*	*gyrB*	*infB*	*pps*	MLST Pattern (No. of Isolates)
*C. sakazakii* 109	O1	44	43	0	1	15	15	80	126	124	56	159	ST256 (4)
						15	67	49	9	14	19	15	ST148 (8)
						10	17	30	59	57	66	83	ST125 (2)
						3	38	120	154	150	15	140	ST654 (1)
						11	8	7	45	8	15	10	ST226 (5)
						1	1	1	1	1	1	1	ST1 (6)
						15	8	39	36	38	38	47	ST58 (5)
						3	8	52	13	21	65	73	ST495 (1)
						3	11	13	18	11	88	104	ST156 (1)
						71	10	68	18	94	92	121	ST188 (1)
						44	69	80	147	96	93	123	ST327 (1)
						15	186	251	298	209	246	362	ST737 (1)
						3	8	13	15	22	20	21	ST50 (2)
						3	8	3	3	18	46	127	ST198 (1)
						3	10	9	275	247	20	51	ST641 (1)
						69	8	13	95	86	105	47	ST219 (1)
						11	8	7	5	8	15	10	ST8 (1)
						16	17	49	32	127	92	245	ST710 (1)
						16	17	49	205	127	92	462 *	ST930 * (1)
	O2	23	21	2	0	3	17	49	68	58	63	65	ST117 (2)
						16	18	120	119	88	73	18	ST308 (4)
						16	8	13	40	15	15	10	ST64 (3)
						48	17	10	69	71	5	81	ST42 (1)
						16	1	19	19	26	5	26	ST22 (1)
						3	12	16	5	16	20	14	ST17 (1)
						20	18	16	10	3	20	27	ST23 (2)
						3	3	3	5	3	3	3	ST3 (3)
						5	1	3	3	5	5	4	ST4 (1)
						16	1	13	39	21	5	21	ST264 (1)
						3	8	37	22	29	36	32	ST31 (1)
						15	14	15	13	22	5	16	ST13 (2)
						10	37	9	3	302	56	126	ST836 (1)
						16	37	9	1	263	92	140	ST938 * (1)
						20	17	306 *	373 *	16	274 *	456 *	ST916 * (1)
	O3	13	11	2	0	11	8	7	5	8	15	10	ST8 (3)
						5	1	3	3	5	5	4	ST4 (3)
						15	8	13	94	99	118	126	ST249 (1)
						3	68	49	88	81	56	96	ST143 (1)
						15	8	13	94	99	98	126	ST199 (1)
						15	1	3	294	15	48	280	ST706 (1)
						15	8	13	94	99	304 *	458 *	ST920 * (1)
						34	36	10	44	43	48	51	ST46 (1)
						16	36	30	58	325 *	303 *	18	ST919 * (1)
	O4	11	11	0	0	3	15	28	22	5	38	19	ST40 (4)
						15	14	15	13	22	5	16	ST13 (2)
						18	17	10	12	18	24	18	ST12 (4)
						15	8	13	94	99	118	126	ST249 (1)
	O7	16	14	2	0	55	14	59	70	70	70	80	ST73 (2)
						19	16	19	41	19	15	23	ST83 (4)
						3	36	52	374	18	90	466	ST921 (1)
						16	36	3	249	58	305 *	4	ST922 * (1)
						3	1	9	378 *	329 *	56	103	ST937 * (1)
						3	36	74	97	86	20	107	ST184 (1)
						3	36	74	97	86	306 *	107	ST924 * (1)
						16	36	3	249	58	36	4	ST587 (1)
						55	1	68	32	191	36	252	ST470 (1)
						15	1	3	32	5	36	190	ST93 (3)
	uncertain	2	0	0	2	3	11	13	18	11	17	13	ST21 (2)
*C.malonaticus* 22	O1	5	4	1	0	12	7	8	8	10	16	43	ST60 (2)
						57	7	64	8	17	16	128	ST211 (1)
						61	7	12	7	189	14	247	ST461 (1)
						10	7	17	8	214	314 *	290	ST936 * (1)
	O2	11	10	0	1	10	7	6	7	9	14	9	ST7 (6)
						10	7	6	99	9	14	9	ST201 (2)
						10	162	67	7	77	204	279	ST567 (1)
						10	13	64	75	239	14	8	ST917 * (2)
	O3	5	4	1	0	61	7	67	7	77	81	95	ST408 (1)
						17	7	17	11	17	22	12	ST11 (3)
						10	13	67	7	131	124	174	ST300 (1)
	uncertain	1	0	0	1	13	7	247	293	77	127	355	ST701 (1)
*C. dublinensis* 19	O1	12	12	0	0	112	41	198	251	211	154	109	ST574 (1)
						94	100	113	133	211	130	282	ST561 (1)
						39	43	45	49	49	52	57	ST77 (3)
						70	20	192	230	84	192	271	ST524 (1)
						94	100	263	133	211	310 *	464 *	ST932 * (1)
						94	100	263	133	211	309 *	463 *	ST931 * (1)
						151	20	70	376 *	327 *	145	36	ST926 * (1)
						67	20	71	91	195	308	461	ST929 * (1)
						67	20	226	92	85	86	460 *	ST928 * (1)
						67	20	307 *	375 *	326 *	214	459 *	ST923 * (1)
	O2	6	6	0	0	233 *	48	34	309	315	27	457 *	ST918 * (1)
						30	46	33	26	36	27	41	ST80 (1)
						58	63	75	76	73	76	108	ST167 (2)
						40	48	133	152	221	312 *	187 *	ST934 * (1)
						58	63	308 *	377 *	73	307 *	108	ST927 * (1)
	uncertain	1	0	1	0	63	20	35	247	119	78	317	ST606 (1)
*C.turicensis* 4	O3	4	4	0	0	68	100	309 *	93	328 *	311 *	465 *	ST933 * (1)
						25	26	22	21	31	37	35	ST35 (1)
						47	5	4	296	116	150	150	ST714 (1)
						47	1	311 *	93	60	313 *	153	ST935 * (1)

## Data Availability

The data used to support the findings of this study can be made available by the corresponding author upon request.

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
