# Peer review of "Cronobacter spp. Isolated from Quick-Frozen Foods in China: Incidence, Genetic Characteristics, and Antibiotic Resistance"

_foods, 2023, doi:10.3390/foods12163019_

Round 1

Reviewer 1 Report

Comments and Suggestions for Authors

Introduction:

- PIF is not a "crucial reservoir". Suggest changing to "common vehicle"

- "can generate extended-spectrum..." rephrase as "can generate" is not correct. Describe the actual mechanism of antimicrobial resistance

- Change "From July 2011 until June 2016" to "From July 2011 to June 2016..."

- Change "Besides fusA gene..." to "In addition, fusA gene..."

Results:

3.1 - change "and no Cronobacter pathogen was detected..." to "and no Cronobacter species were detected..."

Discussion:

"However, we observed a relatively high contamination degree..." - what does "relative" mean here? Is there a rate of contamination that can be used as baseline and/or for comparison?

"Cronobacter spp. infections were treated..." change to "are treated..." In the same paragraph, not clear why authors discuss an "increasing number of resistant strains..." as there are not many that were found the 154 strains tested? Cephalothin is not used to treat this infection.

Change "...are not prudently managed soon." to "...are not carefully managed."

General comments:

1. Throughout the manuscript, bacterial names must be italicized.

2. A very nice study surveilling a large number of studies. Is there anything that can be said about each city, manufacturing practices, etc.? As well, there is no discussion on how the Cronobacter species are getting in, surviving, etc.? This would be very useful.

Comments on the Quality of English Language

The quality of the English language is acceptable. Bacterial names should be italicized. Minor edits/suggestions given.

Reviewer 2 Report

Comments and Suggestions for Authors

In this study, the primary objective was to analyze a substantial number of samples and evaluate the occurrence of Cronobacter in quick-frozen foods from various cities in China.

-The name of microorganisms should be written in italic.

-Keywords and title words should be different form each other.

-“From July 2011until June 2016, a total of 576 quick-frozen food samples were bought from different retail stores and supermarkets in 35 capital cities distributed in 21 provinces and five autonomous regions, two directly controlled municipalities, and two special administrative regions in China” Why did you choose this time zone. 7 years passed. Do you think the results represent todays table?

-The results should have been evaluated with statistical analysis. This is crucial problem in this article.

Comments on the Quality of English Language

Minor editing of English language required.

Round 2

Reviewer 2 Report

Comments and Suggestions for Authors

The article can be acceptable for publishing.

Comments on the Quality of English Language

Minor editing of English language required.

Author Response

Dear  Reviewer,

We would like to thank you for valuable feedback and important comments on our manuscript, and we have edited and revised the English language of the manuscript. Please see the revised manuscript.

We hope the revised manuscript could meet the requirements of reviewer and Foods.